# Using Volatile Organic Compounds to Investigate the Effect of Oral Iron Supplementation on the Human Intestinal Metabolome

**DOI:** 10.3390/molecules25215113

**Published:** 2020-11-03

**Authors:** Ammar Ahmed, Rachael Slater, Stephen Lewis, Chris Probert

**Affiliations:** 1The Henry Wellcome Laboratory, Institute of Systems, Molecular and Integrative Biology, University of Liverpool, Liverpool L69 3BX, UK; Ammar.Ahmed@elht.nhs.uk (A.A.); rsh14@liverpool.ac.uk (R.S.); 2Department of Gastroenterology, University Hospitals Plymouth NHS Trust, Plymouth PL6 8DH, UK; sjl@doctors.org.uk

**Keywords:** iron deficiency anaemia, iron supplementation, volatile organic compounds (VOCs), intestinal metabolome, gut microbiome

## Abstract

Patients with iron deficiency anaemia are treated with oral iron supplementation, which is known to cause gastrointestinal side effects by likely interacting with the gut microbiome. To better study this impact on the microbiome, we investigated oral iron-driven changes in volatile organic compounds (VOCs) in the faecal metabolome. Stool samples from patients with iron deficiency anaemia were collected pre- and post-treatment (*n* = 45 and 32, respectively). Faecal headspace gas analysis was performed by gas chromatography–mass spectrometry and the changes in VOCs determined. We found that the abundance of short-chain fatty acids and esters fell, while aldehydes increased, after treatment. These changes in pre- vs. post-iron VOCs resemble those reported when the gut is inflamed. Our study shows that iron changes the intestinal metabolome, we suggest by altering the structure of the gut microbial community.

## 1. Introduction

Iron is an essential element for numerous metabolic processes, including oxygen transport by haemoglobin and myoglobin [1]. Iron deficiency anaemia (IDA) may arise from inadequate dietary iron intake, malabsorption or blood loss, especially from the gastrointestinal tract [2].

Enteral iron absorption is tightly regulated. Its uptake is dependent on Divalent Metal-Transporter-1, which allows the uptake of iron via the enterocytes in the enteral lumen [1,3]. Other key points are the regulation of transferrin, which binds iron in the circulation; ferrireductases, which facilitate absorption by converting iron from the insoluble ferric (Fe^3+^) form into the soluble ferrous (Fe^2+^) state [1,3]; and hepcidin, an antimicrobial peptide that controls entry of iron into the plasma by binding to and degrading ferroportin, an iron-exporting protein that is very highly expressed on the basolateral membrane of enterocytes [1,2,3]. Under physiological conditions, these series of proteins ensure that enough iron is absorbed, without leading to overload.

Oral iron supplements often exceed the absorptive capacity of the small intestine and the surplus enters the colon where it can lead to gastrointestinal side effects [1,2,4,5] by the generation of free radicals, which damage the epithelium [1,2], and by enhancing the growth of some, but not all, enteric bacteria, leading to dysbiosis [6,7]. These adverse effects can be severe and contribute to the poor compliance with iron therapy.

Volatile organic compounds (VOCs) are carbon-based compounds with a high vapour pressure (and low boiling point) that, consequently, readily enter the gaseous phase. They may contribute to the odours that are associated with faeces and other bodily fluids. Faeces represent the end product of enteric metabolism and digestion, and so changes in faecal VOCs can be used to study changes in the intestinal metabolome and, by extension, the microbiome. Our aim was to explore oral iron-driven changes in the enteral metabolome by comparing the VOC profiles of anaemic patients before and after they had received oral iron supplementation.

## 2. Results

The samples were received as two groups (Table 1).

The median number of VOCs from Group 1 was 73.5 and 77 in the pre- and post-treatment samples, respectively; for Group 2, the values were 68 and 71.5, respectively. Comparison of pre- and post-treatment results in Group 1 found four VOCs that changed significantly in abundance: two aldehydes increased (Table 2). Table 3 summarises the fold change in Group 1, in response to the supplement.

Analysis of the 10 pairs of samples in Group 2 found no compounds that changed to a degree that was statistically significant: this is likely to be a Type 1 error. We looked at the trend in fold change in this cohort (Table 4). There were 27 VOCs that had a greater than two-fold difference between the first and second sample: 6 compounds appeared more and 21 were less abundant; those that increased included pentanal and 2-pentylfuran.

Heatmaps (Figure 1) were generated to illustrate the change in VOCs with treatment, for the two groups. In the first, there is a clear cluster of nine VOCs that increased in abundance with treatment, including four aldehydes (C5–C8), two secondary ketones (C4 and C7), 2-pentylfuran and octen-3-ol. In the second, the six VOCs that increased included two aldehydes (C5 and C7), two secondary ketones (C6 and C7) and 2-pentylfuran.

Box and whisker plots were made to show the change in 2-pentylfuran and pentanal, as these compounds were found to be significantly more abundant in Group 1 and to have greater than two-fold change in abundance in both sets of data in response to treatment (Figure 2). The reduction in esters in Group 1 was of interest as they suggested a change in the metabolism of this class of molecules (Figure 3); however, this was not observed in Group 2.

## 3. Discussion

We investigated the impact of oral iron replacement on the faecal metabolome in two groups of patients. Patients in Group 1 provided unpaired samples of stool, which were taken before and after iron therapy. Patients in Group 2 gave samples twice, enabling a paired analysis. The two groups showed similar results with an increase in faecal aldehydes two months after starting treatment, and there was a reduction in esters.

These changes were more evident, and statistically significant, in Group 1. However, there were similar fold changes in both sets of data (Table 3 and Table 4). Several VOCs, most notably aldehydes, had a greater than two-fold increase in response to iron therapy in the follow-up samples (Table 3 and Table 4). Aldehydes may be generated by lipid peroxidation in response to oxidative stress [8,9]. Inflammation may also cause a change in this class [8]. This finding suggests that non-absorbed iron is damaging to the epithelium; however, changes in the microbiome may be responsible for the increase.

Several ketones derived from secondary alcohols were also increased: such ketones may represent oxidative stress but can be generated by the microbiome. The increase in heptan-2-one in the follow-up samples is consistent with previously reported increases of 2-piperidone and 6-methylheptan-2-one in patients with inflammatory bowel disease [8]. Heptan-2-one may play a part in inhibiting enteric *Escherichia coli* [10].

There was an increase in oct-1-en-3-ol, which is strongly associated with fungi. Oct-1-en-3-ol was found be in increased in patients with active Crohn’s disease [8]. 2-Pentylfuran is also synthesised by fungi [11,12]. It is plausible that there is a change in the mycobiome in response to iron.

Heptanoic acid was markedly reduced in Group 1, following iron therapy. Recent in vitro models have shown that medium chain fatty acids (MCFAs) are able to bind and form ligands with the gamma class of peroxisome proliferator activated receptors (PPAR-γ). MCFAs are produced by intestinal bacteria and may suppress colitis by activating PPAR-γ in macrophages [13].

The abundance of several short-chain fatty (SCFA) and other carboxylic acids was modestly reduced in the follow-up samples. SCFAs are synthesised by bacterial fermentation of dietary fibre [14]. They are important for intestinal health [14,15,16]. The reduction in SCFAs suggests that oral iron causes a reduction in SCFA-producing bacteria. Lee et al. showed that oral iron may suppress *Faecalibacterium prausnitzii* and *Ruminococcus bromii* [17]. Mahalhal et al. reported that iron supplements may suppress Firmicutes, which are a major source of SCFAs [18].

Several fatty/carboxylic esters, including methyl pentanoate, ethyl hexanoate, butyl butanoate and ethyl 2-phenylacetate (Figure 3), were decreased in post-iron samples. A decrease in esters was previously reported in inflammatory bowel disease patients [8]. Faecal esters are likely to be derived from the condensation of fatty acids and have been shown to aid the interactions between fatty acids and gut epithelial cells [19]. A decline in both fatty acids and their esters may be a concomitant process driven by the suppression of intestinal bacteria that produce fatty acids. This contrasts with the increased enteric conversion of fatty acids into their esters, which would have been marked by an increase in faecal esters and a decrease in fatty acids. It should be noted that this change was not observed in the paired samples and it may be an artefact of the unpaired samples (Group 1) or a Type 1 error because Group 2 was smaller.

Esters may have an independent anti-inflammatory role in the gut. For example, oral treatment with branched palmitic acid esters of hydroxy stearic acids has demonstrated a reduction in in vivo colonic T cell activation and in expression of proinflammatory cytokines in mice, along with an in vitro reduction in activation of dendritic cells and in the accompanying proliferation of T cells [20].

The patients who took part in the study all had differing underlying causes of IDA, although they were referred to the clinic with suspected gastrointestinal cancer. Future studies in which the analysis of patients is separated according to different IDA pathologies should be considered. Any ongoing gastrointestinal blood lost would have increased liminal iron and reduced the changes observed in this study. Future studies should look at patients taking iron for other indications (post-operatively or in gynaecological clinics).

## 4. Materials and Methods

### 4.1. Patient and Stool Sample Selection

Patients with iron deficiency anaemia were recruited from a gastroenterology clinic before the cause of the anaemia was diagnosed. Potential recruited patients gave written informed consent before study procedures took place. Patients were treated with standard ferrous sulphate supplementation, up to 200 mg three times/day. Each participant provided two stool samples: the first before commencing iron therapy and the second two months later. Ethical approval for the study was granted by the UK NHS Health Research Authority’s Research Ethics Service (RES) Committee South West—Central Bristol (REC reference 14/SW/1162).

### 4.2. Extraction of VOCs

In order to perform metabolomic analysis, patients’ samples were sent to the University of Liverpool, where they were stored in freezers at either −20 °C or −80 °C before being processed and run through GC-MS (Perkin Elmer Clarus 500, Beaconsfield, UK) apparatus.

With extensive expert technical assistance in the laboratory, all faecal samples were analysed by way of a quadruple GC-MS benchtop system that was used in conjunction with a CombiPAL autosampler (CTC Analytics, Zwingen, Switzerland). The carrier gas used was helium at a very high purity (BOC, Sheffield, UK).

A local laboratory protocol was followed, which was based on the standardised recommendations of GC-MS method optimisation that were proposed by Reade et al. [21]. Every effort was taken to aliquot at least 500 mg of each faecal sample into a vial that had a magnetic cap and a volume of 10 mL. Although this was not possible for every sample, as some samples had limited faecal quantity, all aliquots had a minimum range of 50–100 mg, which was deemed to be sufficient for GC-MS analysis [21]. Thereafter, the samples were incubated at 60 °C for 30 min. Extraction of VOCs from the vial headspace was itself achieved by utilising solvent-free solid phase micro-extraction (SPME) fibres of the CAR-PDMS 85 µm variety (Sigma-Aldrich, Dorset, UK), which were appropriately pre-conditioned prior to use.

### 4.3. Downstream Data Processing and Analysis

After thorough evaluation of each chromatogram produced by the GC-MS, additional data inspection and processing were carried out using the Automated Mass Spectral Deconvolution and Identification System (AMDIS version 2.70, https://amdis.software.informer.com/2.7/) software. This was used in tandem with the US National Institute of Standards and Technology’s (NIST) Mass Spectral Library (version 2.71, (https://www.perkinelmer.com/uk/product/nist-2011-mass-spectral-library-and-software-n6520220)). By manually analysing chromatograms and mass spectra on AMDIS of over 40 of the first samples that were run on the GC-MS, a project-specific library of compounds was created using the NIST database. In addition to taking into account the highest percentage compound match with NIST spectra, VOCs were only included in the library if they had been identified with a minimum match factor of 800.

At this stage, because these samples were blinded, the library was built by incorporating the VOCs found in both pre-iron and post-iron patients. This library eventually comprised over 300 VOCs, which were all named as per the nomenclature standards set by the International Union of Pure and Applied Chemistry (IUPAC) [22]. Once the library was saturated, the iron status (i.e., pre- or post-) of each sample was categorised, and all the samples were collectively analysed together using AMDIS’s batch report function. These data were then further processed and corrected for downstream analysis by using the Metab script package [23], which was utilised within the R (version 3.5, 2018) program.

Subsequently, all statistical analyses were carried out using MetaboAnalyst, a widely used online tool devoted to metabolomic analysis [24]. The principal settings that were utilised included the removal of over 70% of missing metabolite values, though data filtering was not applied. Data were, however, normalised by the median and log-transformed, in addition to being auto-scaled; i.e., mean-centring and then division by the standard deviation of each variable took place. In terms of univariate analysis, fold change analysis and *t*-tests were carried out, with statistical significance being set at a *p*-value less than 0.05. Multivariate analysis centred on principal component analysis (PCA) and partial least squares–discriminant analysis (PLS-DA).

Crucially, in terms of the experimental procedure, VOC extraction was carried out in two batches. Group 1 comprised 35 pre- and 22 post-iron samples that were all from different patients, whereas Group 2 had 10 pre- and 10 post-iron samples each that were all from the same patients (i.e., they were paired samples). Thus, 57 samples were analysed in the first run and a total of 20 samples in the second validation run.

## 5. Conclusions

This study has shown that oral iron replacement is associated with changes in the faecal metabolome. There is an increase in aldehydes, which may be a result of oxidative stress, and a reduction in esters that may reflect an alteration in the microbiome.

## Figures and Tables

**Figure 1 molecules-25-05113-f001:**
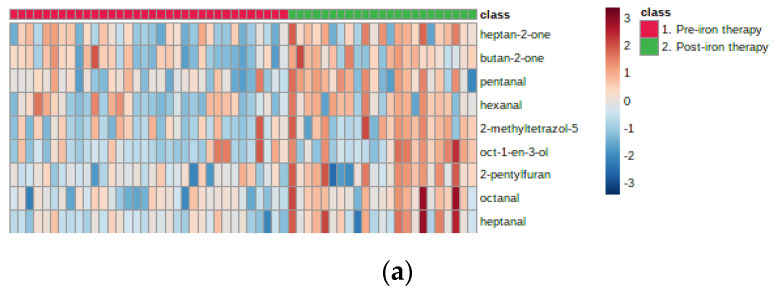
Heatmaps to show the increase in volatile organic compounds (VOCs) with iron therapy in Group 1 (**a**) and Group 2 (**b**).

**Figure 2 molecules-25-05113-f002:**
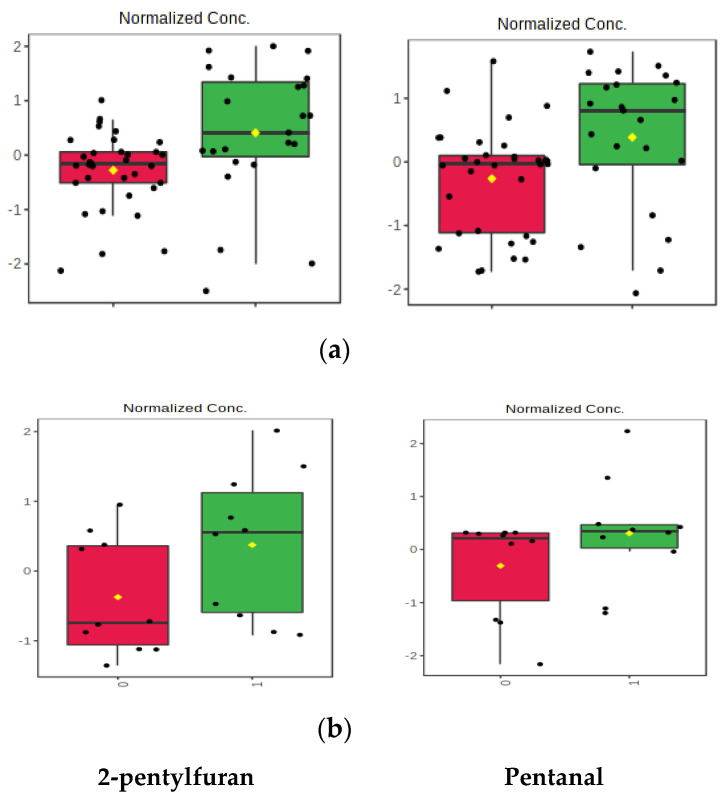
Box and whisker plots showing the change in 2-pentylfuran and pentanal in Group 1 (**a**) and Group 2 (**b**).

**Figure 3 molecules-25-05113-f003:**
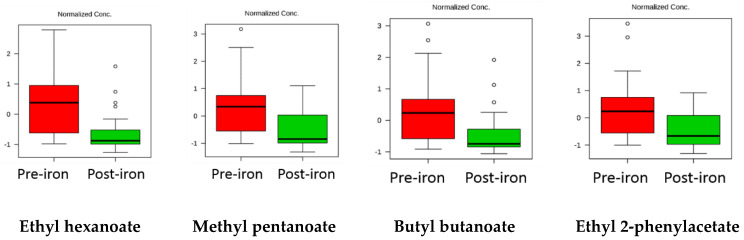
Box and whisker plots show the change in esters in Group 1.

**Table 1 molecules-25-05113-t001:** Summary of age and sex of donors in the two groups.

	Unpaired Samples, Group 1			Paired Samples, Group 2		
	Pre	Post	Total Samples	Pre	Post	Total Samples
Male:Female	21:14	8:14	57	4:6	4:6	20
Mean age (y)	71.4	71.1		69.5	NA	

**Table 2 molecules-25-05113-t002:** Volatiles that changed significantly after iron treatment in Group 1.

	*p*	False Discovery Rate	Trend
Octanal	5.2 × 10^−4^	0.004	Increase
Heptanal	9.8 × 10^−4^	0.019	Increase
Ethyl hexanoate	4 × 10^−4^	0.015	Decrease
2,4-dimethylpentan-3-ol	9.5 × 10^−4^	0.019	Decrease

**Table 3 molecules-25-05113-t003:** Summary of fold change data in Group 1.

VOC That Decreased	Fold Change	VOC That Increased	Fold Change
2,3,5-trimethylpyrazine	15.4	Octanal	7.3
Ethyl 2-phenylacetate	14.9	Heptanal	4.2
Ethyl hexanoate	13.3	2-pentylfuran	4.1
Methyldisulfanylmethane	12.0	Pentanal	3.1
Heptanoic acid	10.7	2-methyltetrazol-5-amine	2.8
4-methylpentanoic acid	8.4	Heptan-2-one	2.7
Methyl pentanoate	8.3	Oct-1-en-3-ol	2.6
Butyl butanoate	7.6		
Ethyl pentanoate	5.9		
Ethyl butanoate	5.6		
Methyl butanoate	4.3		
2,4-dimethylpentan-3-ol	2.9		
Ethenylbenzene	2.7		
1,3-di-tert-butylbenzene	2.6		
Hexanoic acid	2.5		
2-methylbutanoic acid	2.5		
Tetradecane	2.5		
2-methylpropanoic acid	2.4		
5-methyloxolan-2-one	2.4		
2-methylpropanal	2.3		
Ethenyl acetate	2.1		
6,6-dimethyl-2-methylenebicyclo3.1.1heptane	2.1		
Acetic acid	2.1		
1-methyl-3-propan-2-ylbenzene	2.1		

**Table 4 molecules-25-05113-t004:** Summary of fold change data in Group 2.

VOC That Decreased	Fold Change	VOC That Increased	Fold Change
(1*R*,5*S*,6*R*,7*S*,10*R*) 4,10-dimethyl-7-propan-2-yltricyclo(4.4.0.0,5)dec-3-ene	20.1	2-pentylfuran	3.5
3-isopropenyl-1-isopropyl-4-methyl-4-vinylcyclohexene	14.0	methyldisulfanylmethane	3.4
ethyl butanoate	6.4	cyclohexanecarboxylic acid	2.9
butan-1-ol	6.4	hexanal	2.4
4-hydroxy-4-methylpentan-2-one	5.0	pentanal	2.3
4*Z*-4,11,11-trimethyl-8-methylidenebicyclo(7.2.0)undec-4-ene	4.6	pentane-2,3-dione	2.2
1-methyl-4-propan-2-ylcyclohexa-1,4-diene	4.5		
1-methyl-3-propan-2-ylbenzene	4.4		
(5s)- 2-methyl-5-propan-2-ylcyclohexa-1,3-diene	4.3		
5*Z*-2,6,10-trimethyl-1,5,9-undecatriene	4.1		
4-methyl-1-propan-2-ylbicyclo(3.1.0)hex-3-ene	4.0		
7-methyl-3-methylideneocta-1,6-diene	4.0		
6,6-dimethyl-2-methylenebicyclo(3.1.1)heptane	3.5		
4,7,7-trimethylbicyclo(4.1.0)hept-4-ene	3.4		
5-methylheptan-2-one	3.3		
4,6,6-trimethylbicyclo(3.1.1)hept-3-ene	3.1		
2-phenylethanol	2.9		
ethenylbenzene	2.7		
ethylbenzene	2.3		
ethanol	2.2		
1,2-xylene	2.2

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
