# Peer review of "Using Volatile Organic Compounds to Investigate the Effect of Oral Iron Supplementation on the Human Intestinal Metabolome"

_molecules, 2020, doi:10.3390/molecules25215113_

Round 1

Reviewer 1 Report

This paper describes the changing of fecal VOCs in patients treated with oral iron supplementation: the topic is interesting and the used analytical equipment is adequate.

There are some parts that need to be improved.

In the summary is reported “45 patients with iron deficiency anemia were recruited to give pre- and post-treatment stool samples”, but in paragraph “results” you read: “The samples were received as two groups: the first included 35 pre-iron and 22 post-iron samples (unpaired samples provided by 57 patients?), the second contained 10 paired samples pre- and post-iron”. 

The contradiction is evident: you have to clarify how many patients were engaged, and how samples were really collected!

  • In table 1 the acronymous “FDR” is not explained: please report the meaning!
  • In the text, there are no comments or descriptions of data reported in table 2: please include some explanations in the text!
  • In table 2 there are some evident mistakes: Pentanal and 2-pentylfuran are reported twice in the column that includes VOC that increased. Should other products have been included in place of those reported twice?
  • In the text there are no comments about figure 2: you have to explain and comment on the meanings of figure 2. Moreover, why data about 2-pentylfuran and Pentanal are reported while other products that modify in the same way (see table 3) are not shown?
  • In table 3 some names of chemical compounds seem incorrect: please check them and the others.

2-methyl-5-propan-2-ylcyclohexa-1,3-diene?  

4,10-dimethyl-7-propan-2-yltricyclo4.4.0.0,5dec-3-ene?

  • In Figure 3 there are data about Ethyl hexanoate, Methyl pentanoate, Butyl butanoate, and Ethyl 2-phenylacetate. Why these products were chosen and not others? In the text, the reasons for these choices must be included together with the description of data reported in Figure 3.
  • Why all data of pre-treatment were not statistically elaborated on and compare to all post-treatment samples, even if analysed in two different batches? I think that such an analysis could be interesting.
  • The acronymous IBD is not explained: please report the meaning!
  • The acronymous PAHSA is not explained: please report the meaning!

Author Response

This paper describes the changing of fecal VOCs in patients treated with oral iron supplementation: the topic is interesting and the used analytical equipment is adequate. There are some parts that need to be improved.

> Thank you

In the summary is reported “45 patients with iron deficiency anemia were recruited to give pre- and post-treatment stool samples”, but in paragraph “results” you read: “The samples were received as two groups: the first included 35 pre-iron and 22 post-iron samples (unpaired samples provided by 57 patients?), the second contained 10 paired samples pre- and post-iron”. 

The contradiction is evident: you have to clarify how many patients were engaged, and how samples were really collected!

> Thank you. We have adjusted the summary:

"Stool samples, from patients with iron deficiency anaemia, were collected pre- and post-treatment (n = 45 and 32, respectively)."

We have added a new Table to summarise the age and sex of patients recruited in the two groups.

  • In table 1 the acronymous “FDR” is not explained: please report the meaning!

> Thank you, we have removed the abbreviation and replaced it with false discovery rate, in the table now called Table 2.

  • In the text, there are no comments or descriptions of data reported in table 2: please include some explanations in the text!

> We have added the sentence "Table 3 summarises the fold-change in Group 1, in response to the supplement" to line 55. In addition, we have referred the reader back to tables 3 and 4 in the second paragraph of the discussion where fold changes are discussed. 

  • In table 2 there are some evident mistakes: Pentanal and 2-pentylfuran are reported twice in the column that includes VOC that increased. Should other products have been included in place of those reported twice?

> We apologise. This was a cut and paste error: no molecules are missing.

  • In the text there are no comments about figure 2: you have to explain and comment on the meanings of figure 2. Moreover, why data about 2-pentylfuran and Pentanal are reported while other products that modify in the same way (see table 3) are not shown?

> Thank you. On line 63, we draw attention to pentylfuran and pentanal. Later (line 117-119) we have added: "Box and whisker plots were made to show the change in 2-pentylfuran and pentanal as these compounds were found to be significantly more abundant in Group 1 and to have greater than 2-fold change in abundance in both sets of data in response to treatment (Figure 2). 

  • In table 3 some names of chemical compounds seem incorrect: please check them and the others.

2-methyl-5-propan-2-ylcyclohexa-1,3-diene

> (5s) was missing. (5s)- 2-methyl-5-propan-2-ylcyclohexa-1,3-diene. This is the IUPAC name for α-Phellandrene

4,10-dimethyl-7-propan-2-yltricyclo4.4.0.0,5dec-3-ene?

> There were two brackets missing. 4,10-dimethyl-7-propan-2-yltricyclo[4.4.0.01,5]dec-3-ene: this is the IUPAC name for α-Cubebene

> Other missing brackets have been inserted. This is a legacy of the analytic software that is unable to process chemical names that include symbols. so they were removed.  

  • In Figure 3 there are data about Ethyl hexanoate, Methyl pentanoate, Butyl butanoate, and Ethyl 2-phenylacetate. Why these products were chosen and not others? In the text, the reasons for these choices must be included together with the description of data reported in Figure 3.

> Thank you. On line 120 we have added "The reduction in esters, in Group 1 , was of interest as the suggested a change in the metabolism of this class of molecules (Figure 3): however this was not observed in Group 2."

  • Why all data of pre-treatment were not statistically elaborated on and compare to all post-treatment samples, even if analysed in two different batches? I think that such an analysis could be interesting.

> The nature of the two batches were different. Group 1 had pre- and post- treatment samples but many samples were unpaired and were treated as such. Group 2 had paired samples from the same donor: there were analysed as paired samples: the samples in each pair (in Group 2) are likely to more similar to each other than to others in the group: this increases the likelihood that differences found when comparing pairs are real, and not artefacts.

However the two batches should not be combined. If they were combined, then they would have to treated as unpaired samples, when portion of them were paired: this would introduce bias. We have not altered the manuscript.

  • The acronymous IBD is not explained: please report the meaning!

> We have replaced IBD with inflammatory bowel disease.

  • The acronymous PAHSA is not explained: please report the meaning!

> We have explained replace PAHSA in the manuscript. See line 164

Reviewer 2 Report

This is an interesting manuscript assessing changes in the intestinal metabolome resulting from iron therapy.

Specimen collection, and duration of therapy prior to post-treatment specimen collection, are well described.

Presumably the patients who contributed specimens were predominantly women, since that is the population in which iron deficiency is the most prevalent. What was the gender makeup of the patient population, and what was the age distribution?

It does not affect the validity of the results, but many clinicians would now contend that the patients received substantially more iron than was necessary. Recent studies have suggested that ferrous sulfate 200 – 300 mg once daily (or even every other day) produces a comparable results to the traditional three times a day therapy. The authors may find it interesting to do a study later down the line and see if similar changes result from lower dose once daily iron. Some comment on this would enrich the manuscript

On line 30, it would be more clear if the authors said “key points of regulation are…” rather than “key steps are the regulation of…”

It is a little awkward to keep referring to “first group” and “second group”. It would be easier to just refer to the two groups as Group 1 and Group 2.

After line 53, it looks like a paragraph is missing. It is stated that two aldehydes are increased, but the two agents that are decreased are not mentioned. In addition, there is no reference to Table 2.

On lines 57 – 58, it should be stated that no compounds changed to a degree that was statistically significant rather than saying no compounds changed significantly.

Author Response

This is an interesting manuscript assessing changes in the intestinal metabolome resulting from iron therapy. Specimen collection, and duration of therapy prior to post-treatment specimen collection, are well described.

> Thank you

Presumably the patients who contributed specimens were predominantly women, since that is the population in which iron deficiency is the most prevalent. What was the gender makeup of the patient population, and what was the age distribution?

> Thank you. We have added a new Table 1 to summarise age and sex of the participants. In fact, more men than women gave pre-treatment samples. 

It does not affect the validity of the results, but many clinicians would now contend that the patients received substantially more iron than was necessary. Recent studies have suggested that ferrous sulfate 200 – 300 mg once daily (or even every other day) produces a comparable results to the traditional three times a day therapy. The authors may find it interesting to do a study later down the line and see if similar changes result from lower dose once daily iron. Some comment on this would enrich the manuscript

On line 30, it would be more clear if the authors said “key points of regulation are…” rather than “key steps are the regulation of…”

> Thank you. We have made this change.

It is a little awkward to keep referring to “first group” and “second group”. It would be easier to just refer to the two groups as Group 1 and Group 2.

> Thank you, we have made these changes.

After line 53, it looks like a paragraph is missing. It is stated that two aldehydes are increased, but the two agents that are decreased are not mentioned. In addition, there is no reference to Table 2.

> Thank you. We wished to focus on the aldehydes as they were a consistent finding in both parts of the work (new table number 2-4). We comment on this later (line 117-120): "Box and whisker plots were made to show the change in 2-pentylfuran and pentanal as these compounds were found to be significantly more abundant in Group 1 and to have greater than 2-fold change in abundance in both sets of data in response to treatment (Figure 2). The reduction in esters, in Group 1, was of interest as the suggested a change in the metabolism of this class of molecules (Figure 3): however this was not observed in the Group 2.

> We have added a reference to the old table 2 now 3, on line 55/56 "Table 3 summarises the fold-change in Group 1, in response to the supplement."

On lines 57 – 58, it should be stated that no compounds changed to a degree that was statistically significant rather than saying no compounds changed significantly.

> Thank you. We have made this change.

Round 2

Reviewer 1 Report

the Authors have corrected the inaccurate parts and replied to my requirements.